# NONSEPARABLE SYMPLECTIC NEURAL NETWORKS

**Shiying Xiong**[a*], **Yunjin Tong**[a], **Xingzhe He**[a], **Shuqi Yang**[a], **Cheng Yang**[b], **Bo Zhu**[a]

a Dartmouth College, Hanover, NH, United States
b ByteDance AI Lab, Beijing, China

## ABSTRACT

Predicting the behaviors of Hamiltonian systems has been drawing increasing attention in scientific machine learning. However, the vast majority of the literature was focused on predicting separable Hamiltonian systems with their kinematic and potential energy terms being explicitly decoupled while building data-driven paradigms to predict nonseparable Hamiltonian systems that are ubiquitous in fluid dynamics and quantum mechanics were rarely explored. The main computational challenge lies in the effective embedding of symplectic priors to describe the inherently coupled evolution of position and momentum, which typically exhibits intricate dynamics. To solve the problem, we propose a novel neural network architecture, Nonseparable Symplectic Neural Networks (NSSNNs), to uncover and embed the symplectic structure of a nonseparable Hamiltonian system from limited observation data. The enabling mechanics of our approach is an augmented symplectic time integrator to decouple the position and momentum energy terms and facilitate their evolution. We demonstrated the efficacy and versatility of our method by predicting a wide range of Hamiltonian systems, both separable and nonseparable, including chaotic vortical flows. We showed the unique computational merits of our approach to yield long-term, accurate, and robust predictions for large-scale Hamiltonian systems by rigorously enforcing symplectomorphism.

## 1 INTRODUCTION

A Hamiltonian dynamic system refers to a formalism for modeling a physical system exhibiting some specific form of energy conservation during its temporal evolution. A typical example is a pendulum whose total energy (referred to as the system's Hamiltonian) is conserved as a temporally invariant sum of its kinematic energy and potential energy. Mathematically, such energy conservation indicates a specific geometric structure underpinning its time integration, named as a symplectic structure, which further spawns a wide range of numerical time integrators to model Hamiltonian systems. These symplectic time integrators have proven their effectiveness in simulating a variety of energy-conserving dynamics when Hamiltonian expressions are known as a prior. Examples encompass applications in plasma physics (Morrison, 2005), electromagnetics (Li et al., 2019), fluid mechanics (Salmon, 1988), and celestial mechanics (Saari & Xia, 1996), to name a few.

On another front, the emergence of the various machine learning paradigms with their particular focus on uncovering the hidden invariant quantities and their evolutionary structures enable a faithful prediction of Hamiltonian dynamics without knowing its analytical energy expression beforehand. The key mechanics underpinning these learning models lie in a proper embedding of the strong mathematical inductive priors to ensure Hamiltonian conservation in a neural network data flow. Typically, such priors are realized in a variational way or a structured way. For example, in Greydanus et al. (2019), the Hamiltonian conservation is encoded in the loss function. This category of methods does not assume any combinatorial pattern of the energy term and therefore relies on the inherent expressiveness of neural networks to distill the Hamiltonian structure from abundant training datasets (Choudhary et al., 2019). Another category of Hamiltonian networks, which we refer to as structured approaches, implements the conservation law indirectly by embedding a symplectic time integrator (DiPietro et al., 2020; Tong et al., 2020; Chen et al., 2020) or composition of linear, activation, and gradient modules (Jin et al., 2020) into the network architecture.

---

*shiying.xiong@dartmouth.edu

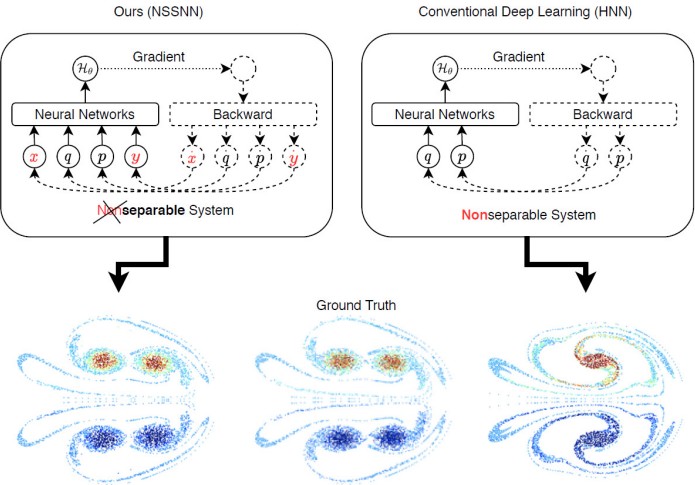

Figure 1: Comparison between NSSNN and HNN regarding the network design and prediction results of a vortex flow example.

.

One of the main limitations of the current structured methods lies in the separable assumption of the Hamiltonian expression. Examples of separable Hamiltonian systems include the pendulum, the Lotka–Volterra (Zhu et al., 2016), the Kepler (Antohe & Gladwell, 2004), and the Hénon–Heiles systems (Zotos, 2015). However, beyond this scope, there exist various nonseparable systems whose Hamiltonian has no explicit expression to decouple the position and momentum energies. Examples include incompressible flows (Suzuki et al., 2007), quantum systems (Bonnabel et al., 2009), rigid body dynamics (Chadaj et al., 2017), charged particle dynamics (Zhang et al., 2016), and nonlinear Schrödinger equation (Brugnano et al., 2018). This nonseparability typically causes chaos and instability, which further complicates the systems' dynamics. Although SympNet in Jin et al. (2020) can be used to learn and predict nonseparable Hamiltonian systems, multiple matrices of the same order with system dimension are needed in the training process of SympNet, resulting in difficulties in generalizing into high-dimensional large-scale N-body problems which are common in a series of nonseparable Hamiltonian systems, such as quantum multibody problems and vortex-particle dynamics problems. Such chaotic and large-scale nature jointly adds shear difficulties for a conventional machine learning model to deliver faithful predictions.

In this paper, we propose an effective machine learning paradigm to predict nonseparable Hamiltonian systems. We build a novel neural network architecture, named nonseparable symplectic neural networks (NSSNNs), to enable accurate and robust predictions of long-term Hamiltonian dynamics based on short-term observation data. Our proposed method belongs to the category of structured network architectures: it intrinsically embeds the symplectomorphism into the network design to strictly preserve the symplectic evolution and further conserves the unknown, nonseparable Hamiltonian energy. The enabling techniques we adopted in our learning framework consist of an augmented symplectic time integrator to asymptotically "decouple" the position and momentum quantities that were nonseparable in their original form. We also introduce the Lagrangian multiplier in the augmented phase space to improve the system's numerical stability. Our network design is motivated by ideas originated from physics (Tao, 2016) and optimization (Boyd et al., 2004). The combination of these mathematical observations and numerical paradigms enables a novel neural network architecture that can drastically enhance both the scale and scope of the current predictions.

We show a motivational example in Figure 1 by comparing our approach with a traditional HNN method (Greydanus et al., 2019) regarding their structural designs and predicting abilities. We refer the readers to Section 6 for a detailed discussion. As shown in Figure 1, the vortices evolved using NSSNN are separated nicely as the ground truth, while the vortices merge together using HNN due to the failure of conserving the symplectic structure of a nonseparable system. The conservative capability of NSSNN springs from our design of the auxiliary variables (red $x$ and $y$) which converts

the original nonseparable system into a higher dimensional quasi-separable system where we can adopt a symplectic integrator.

## 2 RELATED WORKS

**Data-driven physical prediction.** Data-driven approaches have been widely applied in physical systems including fluid mechanics (Brunton et al., 2020), wave physics (Hughes et al., 2019), quantum physics (Sellier et al., 2019), thermodynamics (Hernandez et al., 2020), and material science (Teicherta et al., 2019). Among these different physical systems, data-driven fluid receives increasing attention. We refer the readers to Brunton et al. (2020) for a thorough survey of the fundamental machine learning methodologies as well as their uses for understanding, modeling, optimizing, and controlling fluid flows in experiments and simulations based on training data. One of the motivations of our work is to design a versatile learning approach that can predict complex fluid motions. On another front, many pieces of research focus on incorporating physical priors into the learning framework, e.g., by enforcing incompressibility (Mohan et al., 2020), the Galilean invariance (Ling et al., 2016), quasistatic equilibrium (Geng et al., 2020), the Lagrangian invariance (Cranmer et al., 2020), and Hamiltonian conservation (Hernandez et al., 2020; Greydanus et al., 2019; Jin et al., 2020; Zhong et al., 2020). Here, inspired by the idea of embedding physics priors into neural networks, we aim to accelerate the learning process and improve the accuracy of our model.

**Neural networks for Hamiltonian systems.** Greydanus et al. (2019) introduced Hamiltonian neural networks (HNNs) to conserve the Hamiltonian energy of the system by reformulating the loss function. Inspired by HNN, a series of methods intrinsically embedding a symplectic integrator into the recurrent neural network was proposed, such as SRNN (Chen et al., 2020), TaylorNet (Tong et al., 2020) and SSINN (DiPietro et al., 2020), to solve separable Hamiltonian systems. Combined with graph networks (Sanchez-Gonzalez et al., 2019; Battaglia et al., 2016), these methods were further generalized to large-scale N-body problems induced by interaction force between the particle pairs. Jin et al. (2020) proposed SympNet by directly constructing the symplectic mapping of system variables within neighboring time steps to handle both separable and nonseparable Hamiltonian systems. However, the scale of parameters in SympNet for training $N$ dimensional Hamiltonian system is $O(N^2)$, which makes it hard to be generalized to the high dimensional N-body problems. Our NSSNN overcomes these limitations by devising a new Hamiltonian network architecture that is specifically suited for nonseparable systems (see details in Section 5). In addition, the Hamiltonian-based neural networks can be extended to further applications. Toth et al. (2020) developed the Hamiltonian Generative Network (HGN) to learn Hamiltonian dynamics from high-dimensional observations (such as images). Moreover, Zhong et al. (2020) introduced Symplectic ODE-Net (SymODEN), which adds an external control term to the standard Hamiltonian dynamics.

## 3 FRAMEWORK

### 3.1 AUGMENTED HAMILTONIAN EQUATION

We start by considering a Hamiltonian system with $N$ pairs of canonical coordinates (i.e. $N$ generalized positions and $N$ generalized momentum). The time evolution of canonical coordinates is governed by the symplectic gradient of the Hamiltonian (Hand & Finch, 2008). Specifically, the time evolution of the system is governed by Hamilton's equations as

$$\frac{\mathrm{d}\boldsymbol{q}}{\mathrm{d}t} = \frac{\partial \mathcal{H}}{\partial \boldsymbol{p}}, \quad \frac{\mathrm{d}\boldsymbol{p}}{\mathrm{d}t} = -\frac{\partial \mathcal{H}}{\partial \boldsymbol{q}}, \tag{1}$$

with the initial condition $(\boldsymbol{q}, \boldsymbol{p})|_{t=t_0} = (\boldsymbol{q}_0, \boldsymbol{p}_0)$. In a general setting, $\boldsymbol{q} = (q_1, q_2, \cdots, q_N)$ represents the positions and $\boldsymbol{p} = (p_1, p_2, ...p_N)$ denotes their momentum. Function $\mathcal{H} = \mathcal{H}(\boldsymbol{q}, \boldsymbol{p})$ is the Hamiltonian, which corresponds to the total energy of the system. An important feature of Hamilton's equations is its symplectomorphism (see Appendix B for a detailed overview).

The symplectic structure underpinning our proposed network architecture draws inspirations from the original research of Tao (2016) in computational physics. In Tao (2016), a generic, high-order, explicit and symplectic time integrator was proposed to solve (1) of an arbitrary separable and nonseparable

Hamiltonian $\mathcal{H}$. This is implemented by considering an augmented Hamiltonian

$$\overline{\mathcal{H}}(\boldsymbol{q}, \boldsymbol{p}, \boldsymbol{x}, \boldsymbol{y}) := \mathcal{H}_A + \mathcal{H}_B + \omega \mathcal{H}_C \tag{2}$$

with

$$\mathcal{H}_A = \mathcal{H}(\boldsymbol{q}, \boldsymbol{y}), \;\; \mathcal{H}_B = \mathcal{H}(\boldsymbol{x}, \boldsymbol{p}), \;\; \mathcal{H}_C = \frac{1}{2}\left(\|\boldsymbol{q} - \boldsymbol{x}\|_2^2 + \|\boldsymbol{p} - \boldsymbol{y}\|_2^2\right) \tag{3}$$

in an extended phase space with symplectic two form $\mathrm{d}\boldsymbol{q} \wedge \mathrm{d}\boldsymbol{p} + \mathrm{d}\boldsymbol{x} \wedge \mathrm{d}\boldsymbol{y}$, where $\omega$ is a constant that controls the binding of the original system and the artificial restraint.

Notice that the Hamilton's equations for $\overline{\mathcal{H}}$

$$\begin{cases} \dfrac{\mathrm{d}\boldsymbol{q}}{\mathrm{d}t} = \dfrac{\partial \overline{\mathcal{H}}}{\partial \boldsymbol{p}} = \dfrac{\partial \mathcal{H}(\boldsymbol{x}, \boldsymbol{p})}{\partial \boldsymbol{p}} + \omega(\boldsymbol{p} - \boldsymbol{y}), \\[2mm] \dfrac{\mathrm{d}\boldsymbol{p}}{\mathrm{d}t} = -\dfrac{\partial \overline{\mathcal{H}}}{\partial \boldsymbol{q}} = -\dfrac{\partial \mathcal{H}(\boldsymbol{q}, \boldsymbol{y})}{\partial \boldsymbol{q}} - \omega(\boldsymbol{q} - \boldsymbol{x}), \\[2mm] \dfrac{\mathrm{d}\boldsymbol{x}}{\mathrm{d}t} = \dfrac{\partial \overline{\mathcal{H}}}{\partial \boldsymbol{y}} = \dfrac{\partial \mathcal{H}(\boldsymbol{q}, \boldsymbol{y})}{\partial \boldsymbol{y}} - \omega(\boldsymbol{p} - \boldsymbol{y}), \\[2mm] \dfrac{\mathrm{d}\boldsymbol{y}}{\mathrm{d}t} = -\dfrac{\partial \overline{\mathcal{H}}}{\partial \boldsymbol{x}} = -\dfrac{\partial \mathcal{H}(\boldsymbol{x}, \boldsymbol{p})}{\partial \boldsymbol{x}} + \omega(\boldsymbol{q} - \boldsymbol{x}), \end{cases} \tag{4}$$

with the initial condition $(\boldsymbol{q}, \boldsymbol{p}, \boldsymbol{x}, \boldsymbol{y})|_{t=t_0} = (\boldsymbol{q}_0, \boldsymbol{p}_0, \boldsymbol{q}_0, \boldsymbol{p}_0)$ have the same exact solution as (1) in the sense that $(\boldsymbol{q}, \boldsymbol{p}, \boldsymbol{x}, \boldsymbol{y}) = (\boldsymbol{q}, \boldsymbol{p}, \boldsymbol{q}, \boldsymbol{p})$. Hence, we can get the solution of (1) by solving (4). Furthermore, it is possible to construct high-order symplectic integrators for $\overline{\mathcal{H}}$ in (4) with explicit updates. Our model aims to learn the dynamical evolution of $(\boldsymbol{q}, \boldsymbol{p})$ in (1) by embedding (4) into the framework of NeuralODE (Chen et al., 2018). The coefficient $\omega$ acts as a regularizer, which stabilizes the numerical results (see Section 4).

### 3.2 NONSEPARABLE HAMILTONIAN NEURAL NETWORK

We learn the nonseparable Hamiltonian dynamics (1) by constructing an augmented system (4), from which we can obtain the energy function $\mathcal{H}(\boldsymbol{q}, \boldsymbol{p})$ by training the neural network $\mathcal{H}_\theta(\boldsymbol{q}, \boldsymbol{p})$ with parameter $\boldsymbol{\theta}$ and calculate the gradient $\boldsymbol{\nabla}\mathcal{H}_\theta(\boldsymbol{q}, \boldsymbol{p})$ by taking the in-graph gradient. For the constructed network $\mathcal{H}_\theta(\boldsymbol{q}, \boldsymbol{p})$, we integrate (4) by using the second-order symplectic integrator (Tao, 2016). Specifically, we will have an input layer $(\boldsymbol{q}, \boldsymbol{p}, \boldsymbol{x}, \boldsymbol{y}) = (\boldsymbol{q}_0, \boldsymbol{p}_0, \boldsymbol{q}_0, \boldsymbol{p}_0)$ at $t = t_0$ and an output layer $(\boldsymbol{q}, \boldsymbol{p}, \boldsymbol{x}, \boldsymbol{y}) = (\boldsymbol{q}_n, \boldsymbol{p}_n, \boldsymbol{x}_n, \boldsymbol{y}_n)$ at $t = t_0 + n\mathrm{d}t$.

---

**Algorithm 1** Integrate (4) by using the second-order symplectic integrator

**Input:** $\boldsymbol{q}_0, \boldsymbol{p}_0, t_0, t, \mathrm{d}t;\;\; \phi_1^\delta, \phi_2^\delta,$ and $\phi_3^\delta$ in (5);
**Output:** $(\hat{\boldsymbol{q}}, \hat{\boldsymbol{p}}, \hat{\boldsymbol{x}}, \hat{\boldsymbol{y}}) = (\boldsymbol{q}_n, \boldsymbol{p}_n, \boldsymbol{x}_n, \boldsymbol{y}_n)$

1   $(\boldsymbol{q}_0, \boldsymbol{p}_0, \boldsymbol{x}_0, \boldsymbol{y}_0) = (\boldsymbol{q}_0, \boldsymbol{p}_0, \boldsymbol{q}_0, \boldsymbol{p}_0)$   $n = $ floor$[(t - t_0)/\mathrm{d}t]$ **for** $i = 1 \rightarrow n$ **do**

2    $(\boldsymbol{q}_i, \boldsymbol{p}_i, \boldsymbol{x}_i, \boldsymbol{y}_i) = \phi_1^{\mathrm{d}t/2} \circ \phi_2^{\mathrm{d}t/2} \circ \phi_3^{\mathrm{d}t} \circ \phi_2^{\mathrm{d}t/2} \circ$ $\phi_1^{\mathrm{d}t/2} \circ (\boldsymbol{q}_{i-1}, \boldsymbol{p}_{i-1}, \boldsymbol{x}_{i-1}, \boldsymbol{y}_{i-1});$

3 **end**

---

The recursive relations of $(\boldsymbol{q}_i, \boldsymbol{p}_i, \boldsymbol{x}_i, \boldsymbol{y}_i), i = 1, 2, \cdots, n$, can be expressed by the algorithm 1 (also see Figure 8 in Appendix A). The input functions $\phi_1^\delta(\boldsymbol{q}, \boldsymbol{p}, \boldsymbol{x}, \boldsymbol{y})$, $\phi_2^\delta(\boldsymbol{q}, \boldsymbol{p}, \boldsymbol{x}, \boldsymbol{y})$, and $\phi_3^\delta(\boldsymbol{q}, \boldsymbol{p}, \boldsymbol{x}, \boldsymbol{y})$ in algorithm 1 are

$$\begin{bmatrix} \boldsymbol{q} \\ \boldsymbol{p} - \delta[\partial\mathcal{H}_\theta(\boldsymbol{q}, \boldsymbol{y})/\partial\boldsymbol{q}] \\ \boldsymbol{x} + \delta[\partial\mathcal{H}_\theta(\boldsymbol{q}, \boldsymbol{y})/\partial\boldsymbol{p}] \\ \boldsymbol{y} \end{bmatrix}, \;\; \begin{bmatrix} \boldsymbol{q} + \delta[\partial\mathcal{H}_\theta(\boldsymbol{x}, \boldsymbol{p})/\partial\boldsymbol{p}] \\ \boldsymbol{p} \\ \boldsymbol{x} \\ \boldsymbol{y} - \delta[\partial\mathcal{H}_\theta(\boldsymbol{x}, \boldsymbol{p})/\partial\boldsymbol{q}] \end{bmatrix}, \text{ and } \frac{1}{2}\begin{bmatrix} \begin{pmatrix} \boldsymbol{q} + \boldsymbol{x} \\ \boldsymbol{p} + \boldsymbol{y} \end{pmatrix} + \boldsymbol{R}^\delta\begin{pmatrix} \boldsymbol{q} - \boldsymbol{x} \\ \boldsymbol{p} - \boldsymbol{y} \end{pmatrix} \\ \begin{pmatrix} \boldsymbol{q} + \boldsymbol{x} \\ \boldsymbol{p} + \boldsymbol{y} \end{pmatrix} - \boldsymbol{R}^\delta\begin{pmatrix} \boldsymbol{q} - \boldsymbol{x} \\ \boldsymbol{p} - \boldsymbol{y} \end{pmatrix} \end{bmatrix}, \tag{5}$$

respectively. Here

$$\boldsymbol{R}^\delta := \begin{bmatrix} \cos(2\omega\delta)\boldsymbol{I} & \sin(2\omega\delta)\boldsymbol{I} \\ -\sin(2\omega\delta)\boldsymbol{I} & \cos(2\omega\delta)\boldsymbol{I} \end{bmatrix}, \;\; \text{where } \boldsymbol{I} \text{ is a identity matrix.} \tag{6}$$

We remark that $\boldsymbol{x}$ and $\boldsymbol{y}$ are just auxiliary variables, which are theoretically equal to $\boldsymbol{q}$ and $\boldsymbol{p}$. Therefore, we can use the data set of $(\boldsymbol{q}, \boldsymbol{p})$ to construct the data set containing variables $(\boldsymbol{q}, \boldsymbol{p}, \boldsymbol{x}, \boldsymbol{y})$. In addition, by constructing the network $\mathcal{H}_\theta$, we show that theorem B.1 in Appendix B holds, so the networks $\phi_1^\delta, \phi_2^\delta$, and $\phi_3^\delta$ in (5) preserve the symplectic structure of the system. Suppose that $\Phi_1$ and $\Phi_2$ are two symplectomorphisms. Then, it is easy to show that their composite map $\Phi_2 \circ \Phi_1$ is also symplectomorphism due to the chain rule. Thus, the symplectomorphism of algorithm 1 can be guaranteed by the theorems B.1.

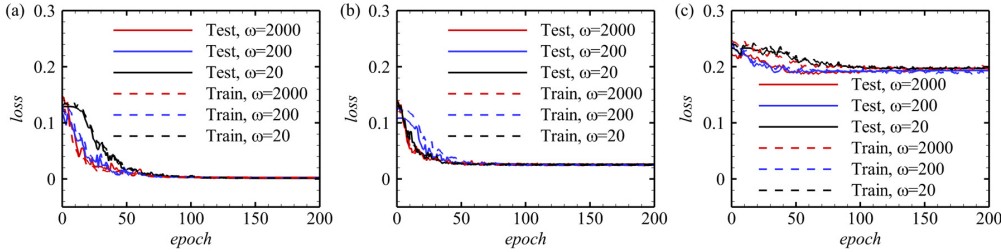

Figure 2: Comparisons of training and validation losses with different $\omega$ in the training process. The solid and dashed lines represent training and validation losses respectively. The networks are trained by (a) the clean dataset, (b) the dataset with noise $\sim 0.05U(-1,1)$, and (c) the dataset with noise $\sim 0.4U(-1,1)$.

## 4    TRAINING SETTINGS AND ABLATION TESTS

We use 6 linear layers with hidden size 64 to model $\mathcal{H}_\theta$, all of which are followed by a Sigmoid activation function except the last one. The derivatives $\partial\mathcal{H}_\theta/\partial p, \partial\mathcal{H}_\theta/\partial q, \partial\mathcal{H}_\theta/\partial x, \partial\mathcal{H}_\theta/\partial y$ are all obtained by automatic differentiation in Pytorch (Paszke et al., 2019). The weights of the linear layers are initialized by Xavier initializaiton (Glorot & Bengio, 2010).

We generate the dataset for training and validation using high-precision numerical solver (Tao, 2016), where the ratio of training and validation datasets is $9:1$. We set the dataset $(q_0^j, p_0^j)$ as the start input and $(q^j, p^j)$ as the target with $j = 1, 2, \cdots, N_s$, and the time span between $(q_0^j, p_0^j)$ and $(q^j, p^j)$ is $T_{train}$. Feeding $(q_0, p_0) = (q_0^j, p_0^j)$, $t_0 = 0$, $t = T_{train}$, and time step d$t$ in Algorithm 1 to get the predicted variables $(\hat{q}^j, \hat{p}^j, \hat{x}^j, \hat{y}^j)$. Accordingly, the loss function is defined as

$$\mathcal{L}_{NSSNN} = \frac{1}{N_b} \sum_{j=1}^{N_b} \|q^{(j)} - \hat{q}^{(j)}\|_1 + \|p^{(j)} - \hat{p}^{(j)}\|_1 + \|q^{(j)} - \hat{x}^{(j)}\|_1 + \|p^{(j)} - \hat{y}^{(j)}\|_1, \quad (7)$$

where $N_b = 512$ is the batch size of the training samples. We use the Adam optimizer (Kingma & Ba, 2015) with learning rate 0.05. The learning rate is multiplied by 0.8 for every 10 epoches.

Taking system $\mathcal{H}(q,p) = 0.5(q^2 + 1)(p^2 + 1)$ as an example, we carry out a series of ablation tests based on our constructed networks. Normally, we set the time span, time step and dateset size as $T = 0.01$, d$t = 0.01$ and $N_s = 1280$.

The choice of $\omega$ in (4) is largely flexible since NSSNN is not sensitive to the parameter $\omega$ when it is larger than a certain threshold. Figure 2 shows the training and validation losses with different $\omega$ in the network trained by clean and noise datasets. Though the convergence rates are slightly different in a small scope, the examples with various $\omega$ are able to converge to the same size of training and validation losses. Here, we set $\omega = 2000$, but $\omega$ can be smaller than 2000. The only requirement for picking $\omega$ is that it has to be larger than $O(10)$, which is detailed in Appendix C.

We pick the $L1$ loss function to train our network due to its better performance. Figure 3 compares the validation losses with different training loss functions in the network trained by clean and noise datasets. Figure 3(a) shows that either the network trained by $L1$ or MSE with a clean dataset can converge to a small validation loss, but the network trained by $L1$ loss converges relatively faster. Figures 3(b) and 3(c) both show that the network trained by $L1$ with noise dataset can converge to a smaller validation loss. In addition, we already introduced a regularization term in the symplectic integrator embedded in the network; thus, there is no need to add the regularization term in the loss function.

The integral time step in the sympletic integrator is a vital parameter, and the choice of d$t$ largely depends on the time span $T_{train}$. Figure 4 compares the validation losses generated by various integral time steps d$t$ based on fixed dataset time spans $T_{train} = 0.01, 0.1$ and $0.2$ respectively in the training process. The validation loss converges to a similar degree with various d$t$ based on fixed $T_{train} = 0.01$ and $T_{train} = 0.1$ in 4(a) and (b), while it increases significantly as d$t$ increases based

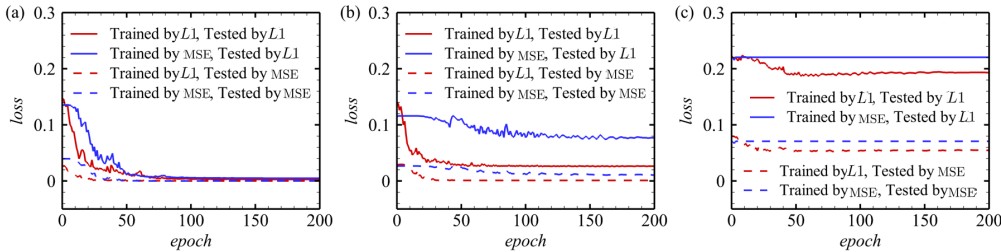

Figure 3: Comparisons of validation losses with different training loss functions in the training process. The red solid and dashed lines represent the networks trained by $L1$ loss function and validated by $L1$ and MSE respectively; the blue solid and dashed lines represent the networks trained by MSE and validated by $L1$ and MSE respectively. The networks are trained by (a) the clean dataset, (b) the dataset with noise $\sim 0.05U(-1, 1)$, and (c) the dataset with noise $\sim 0.4U(-1, 1)$.

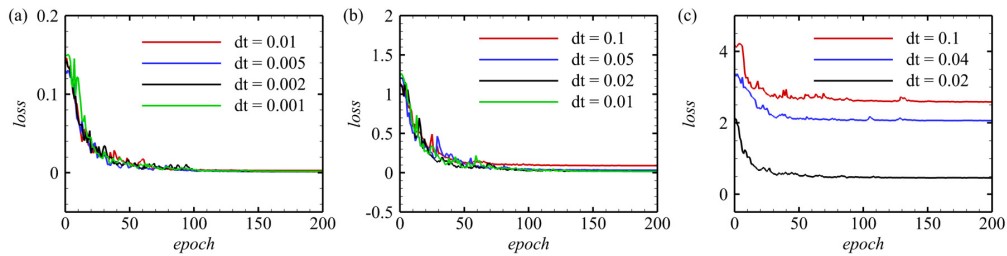

Figure 4: Comparisons of validation losses with different d$t$ in the training process. (a), (b), and (c) are trained based on different time spans $T_{train} = 0.01$, $0.1$, and $0.2$, respectively.

on fixed $T_{train} = 0.02$ in 4(c). Thus, we should take relatively small d$t$ for the dataset with larger time span $T_{train}$

## 5 COMPARISONS WITH OTHER METHODS

### 5.1 METHODOLOGIES

We compare our method with other recently proposed methods, such as HNN (Greydanus et al., 2019), NeuralODE (Chen et al., 2018), TaylorNet (Tong et al., 2020), SSINN (DiPietro et al., 2020), SRNN (Chen et al., 2020), and SympNet (Jin et al., 2020). There are several features distinguishing our method from others, as shown in Table 1. HNN first enforces conservative features of a Hamiltonian system by reformulating its loss function, which incurs two main shortcomings. On the one hand, it requires the temporal derivatives of the momentum and the position of the systems to calculate the loss function, which is difficult to obtain from real-world systems. On the other hand, HNN doesn't strictly preserve the symplectic structure, because its symplectomorphism is realized by its loss function rather than its intrinsic network architecture. NeuralODE successfully bypasses the time derivatives of the datasets by incorporating an integrator solver into the network architecture.

Embedding the Hamiltonian prior into the NeuralODE, a series of methods are proposed, such as SRNN, SSINN, and TaylorNet, to predict the continuous trajectory of system variables; however, presently these methods are only designed to solve separable Hamiltonian systems. Instead of updating the continuous dynamics by integrating the neural networks in NeuralODE, SympNet adopts a symplectomorphism composed of well-designed both linear and non-linear matrices to intrinsically map the system variables within neighboring time steps. However, the parameters scale in the matrix map for training $N$ dimensional Hamiltonian system in SympNet is $O(N^2)$, which makes it hard to generalize to the high dimensional N-body problems. For example, in Section 6, we predict the dynamic evolution of 6000 vortex particles, which is challenging for the training process of the SympNet on the level of $O(6000^2)$.

Table 1: Comparison between HNN (Greydanus et al., 2019), NeuralODE (Chen et al., 2018), TaylorNet (Tong et al., 2020), SSINN (DiPietro et al., 2020), SRNN (Chen et al., 2020), SympNet (Jin et al., 2020), and NSSNN. ✓ represents the method preserves such property.

| Methods | NSSNN | HNN | NeuralODE | TaylorNet | SSINN | SRNN | SympNet |
|---|---|---|---|---|---|---|---|
| **Solve nonseparable systems** | ✓ | ✓ | ✓ | | | | ✓ |
| Solve separable systems | ✓ | ✓ | ✓ | ✓ | ✓ | ✓ | ✓ |
| Preserve symplectic structure | ✓ | Partially | | ✓ | Partially | ✓ | ✓ |
| Utilize continuous dynamics | ✓ | | ✓ | ✓ | ✓ | ✓ | ✓ |
| No need for derivatives in dataset | ✓ | | ✓ | ✓ | ✓ | ✓ | ✓ |
| Long-term predictability | ✓ | Partially | | ✓ | ✓ | ✓ | ✓ |
| Extend to N-body system | ✓ | ✓ | ✓ | ✓ | | ✓ | |

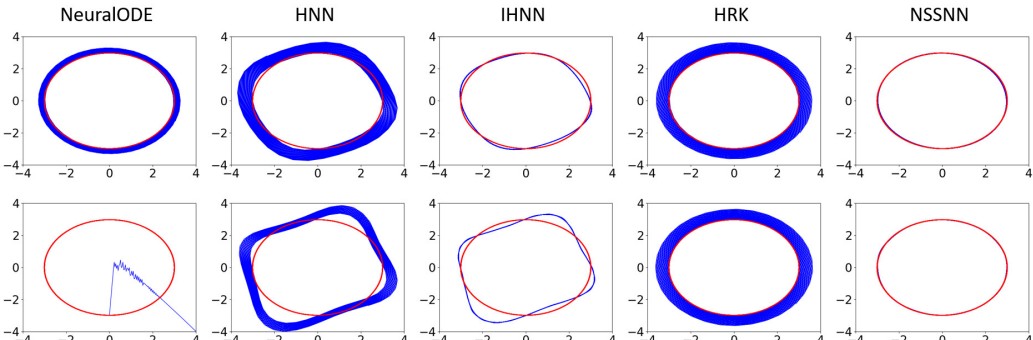

Figure 5: Comparison of prediction results of $(q, p)$ for the spring system $\mathcal{H} = 0.5(q^2 + p^2)$ from $t = 0$ to $t = 200$ with $(q_0, p_0) = (0, -3)$. The time span of the datasets are $T_{train} = 0.4$ (first row) and $T_{train} = 1$ (second row). The five columns are five different methods NeuralODE, HNN, IHNN, HRK, and NSSNN, respectively. The red line denotes the ground truth; the blue line denotes the prediction, which are perfectly overlapping in NSSNN. The prediction ability of HNN and IHNN improves significantly with the decreasing of $T_{train}$ of the dataset which however may be hard to obtain in the actual experimental measurements.

NSSNN overcomes the weaknesses mentioned above. Under the framework of NeuralODE, NSSNN utilizes continuously-defined dynamics in the neural networks, which gives it the capability to learn the continuous-time evolution of dynamical systems. Based on Tao (2016), NSSNN embeds the symplectic prior into the nonseparable symplectic integrator to ensure the strict symplectomorphism, thereby guaranteeing the property of long-term predictability. In addition, unlike SympNet, NSSNN is highly flexible and can be generalized to high dimensional N-body problems by involving the interaction networks (Sanchez-Gonzalez et al., 2019), which will be further discussed in Section 6.

## 5.2 EXPERIMENTS

We compare five implementations that learn and predict Hamiltonian systems. The first one is NeuralODE, which trains the system by embedding the network $\boldsymbol{f}_\theta \to (\mathrm{d}\boldsymbol{q}/\mathrm{d}t, \mathrm{d}\boldsymbol{p}/\mathrm{d}t)$ into the Runge-Kutta (RK) integrator. The other four, however, achieve the goal by fitting the Hamiltonian $\mathcal{H}_\theta \to \mathcal{H}$ based on (1). Specifically, HNN trains the network with the constraints of the Hamiltonian symplectic gradient along with the time derivative of system variables and then embeds the well-trained $\mathcal{H}_\theta$ into the RK integrator for predicting the system. The third and fourth implementations are ablation tests. One of them is improved HNN (IHNN), which embeds the well-trained $\mathcal{H}_\theta$ into the nonseparable symplectic integrator (Tao's integrator) for predicting. The other is to directly embed $\mathcal{H}_\theta$ into the RK integrator for training, which we call HRK. The fifth method is NSSNN, which embeds $\mathcal{H}_\theta$ into the nonseparable symplectic integrator for training.

For fair comparison, we adopt the same network structure (except that the dimension of output layer in NeuralODE is two times larger than that in the other four), the same $L1$ loss function and same

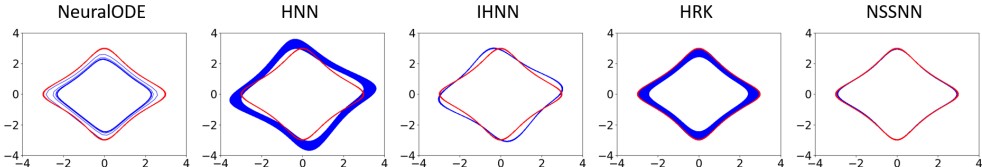

Figure 6: Comparison of prediction results of $(q, p)$ for the Tao's system $\mathcal{H} = 0.5(q^2 + 1)(p^2 + 1)$ from $t = 0$ to $t = 20000$ with $(q_0, p_0) = (0, -3)$. The network is trained by the dataset with noise $\sim 0.05U(-1, 1)$, and the time span of the dataset is $T_{train} = 0.2$. The red line denotes the ground truth; the blue line denotes the prediction, which are perfectly overlapping in NSSNN.

size of the dataset, and the precision of all integral schemes is second order, and the other parameters keep consistent with the one in Section 4. The time derivative in the dataset for training HNN and IHNN is obtained by the first difference method

$$\frac{\mathrm{d}\boldsymbol{q}}{\mathrm{d}t} \approx \frac{\boldsymbol{q}(T_{train}) - \boldsymbol{q}(0)}{T_{train}} \quad \text{and} \quad \frac{\mathrm{d}\boldsymbol{p}}{\mathrm{d}t} \approx \frac{\boldsymbol{p}(T_{train}) - \boldsymbol{q}(0)}{T_{train}}. \tag{8}$$

Figure 5 demonstrates the differences between the five methods using a spring system $\mathcal{H} = 0.5(q^2 + p^2)$ with different time span $T_{train} = 0.4, 1$ and same time step $\mathrm{d}t = 0.2$. We can see that by introducing the nonseparable symplectic integrator into the prediction of the Hamiltonian system, NSSNN has a stronger long-term predicting ability than all the other methods. In addition, the prediction of HNN and IHNN lies in the dataset with time derivative; consequently, it will lead to a larger error when the given time span $T_{train}$ is large.

Moreover, the datasets obtained by (11) in HNN and IHNN are sensitive to noise. Figure 6 compares the predictions of $(q, p)$ for the system $\mathcal{H} = 0.5(q^2 + 1)(p^2 + 1)$, where the network is trained by the dataset with noise $\sim 0.05U(-1, 1)$, along with time span $T_{train} = 0.2$ and time step $\mathrm{d}t = 0.02$. Under the condition with noise, NSSNN still performs well compared with other methods. Also, we compare the convergent error of a series of Hamiltonian systems with different $\mathcal{H}$ trained with noisy data in Appendix D, which generally shows better robustness than HNN does.

## 6 MODELING VORTEX DYNAMICS OF MULTI-PARTICLE SYSTEM

For two-dimensional vortex particle systems, the dynamical equations of particle positions $(x_j, y_j)$, $j = 1, 2, \cdots, N_v$ with particle strengths $\Gamma_j$ can be written in the generalized Hamiltonian form as

$$\Gamma_j \frac{\mathrm{d}x_j}{\mathrm{d}t} = -\frac{\partial \mathcal{H}^p}{\partial y_j}, \quad \Gamma_j \frac{\mathrm{d}y_j}{\mathrm{d}t} = \frac{\partial \mathcal{H}^p}{\partial x_j}, \quad \text{with} \quad \mathcal{H}^p = \frac{1}{4\pi} \sum_{j,k=1}^{N_v} \Gamma_j \Gamma_k \log(|x_j - x_k|). \tag{9}$$

By including the given particle strengths $\Gamma_j$ in Algorithm 1, we can still adopt the method mentioned above to learn the Hamiltonian in (9) when there are fewer particles. However, considering a system with $N_v \gg 2$ particles, the cost to collect training data from all $N_v$ particles might be high, and the training process can be time-consuming. Thus, instead of collecting information from all $N_v$ particles to train our model, we only use data collected from two bodies as training data to make predictions of the dynamics of $N_v$ particles.

Specifically, we assume the interactive models between particle pairs with unit particle strengths $\Gamma_j = 1$ are the same, and their corresponding Hamiltonian can be represented as network $\hat{\mathcal{H}}_\theta(\boldsymbol{x}_j, \boldsymbol{x}_k)$, based on which the corresponding Hamiltonian of $N_v$ particles can be written as (Battaglia et al., 2016; Sanchez-Gonzalez et al., 2019)

$$\mathcal{H}_\theta^p = \sum_{i,j=1}^{N_v} \Gamma_j \Gamma_k \hat{\mathcal{H}}_\theta(\boldsymbol{x}_j, \boldsymbol{x}_k). \tag{10}$$

We embed (10) into the symplectic integrator that includes $\Gamma_j$ to obtain the final network architecture.

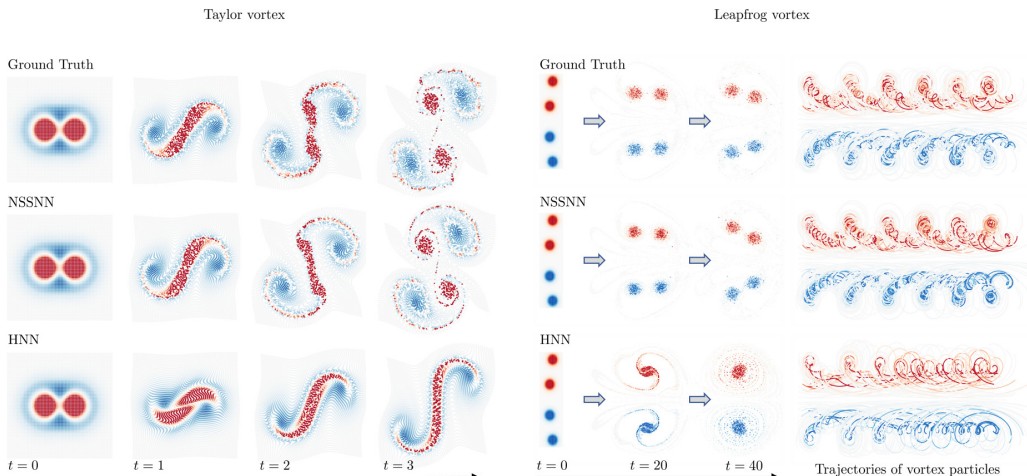

Figure 7: Taylor and Leapfrog vortex. We generate results of Taylor vortex and Leapfrop vortex using NSSNN and HNN, and compare them with the ground truth. 6000 vortex elements are used with corresponding initial vorticity conditions of Taylor vortex and Leapfrop vortex.

The setup of the multi-particle problem is similar to the previous problems. The training time span is $T_{train} = 0.01$ while the prediction period can be up to $T_{predict} = 40$. We use 2048 clean data samples to train our model. The training process takes about 100 epochs for the loss to converge. In Figure 7, we use our trained model to predict the dynamics of 6000-particle systems, including Taylor and Leapfrog vortices. We generate results of Taylor vortex and Leapfrop vortex using NSSNN and HNN and compare them with the ground truth. Vortex elements are used with corresponding initial vorticity conditions of Taylor vortex and Leapfrop vortex (Qu et al., 2019). The difficulty of the numerical modeling of these two systems lies in the separation of different dynamical vortices instead of having them merging into a bigger structure. In both cases, the vortices evolved using NSSNN are separated nicely as the ground truth shows, while the vortices merge together using HNN.

## 7  LIMITATIONS

The network with the embedded integrator is often more time-consuming to train than the one based on the dataset with time derivative. For example, the ratio of training time of the methods HNN and NSSNN is $1 : 3$ when $dt = T_{train}$, and the training time of the recurrent networks further increases with the decreasing of $dt$. Although a smaller $dt$ often has higher discretization accuracy, there is a tradeoff between training cost and predicting accuracy. Additionally, a smaller $dt$ may potentially cause gradient explosion. In this case, we may want to use the adjoint method instead. Another limitation lies in the assumption that the symplectic structure is conserved. In real-world systems, there could be dissipation that makes this assumption unsatisfied.

## 8  CONCLUSIONS

We incorporate a classic ideal that maps a nonseparable system to a higher dimensional space making it quasi-separable to construct symplectic networks. With the intrinsic symplectic structure, NSSNN possesses many benefits compared with other methods. In particular, NSSNN is the first method that can learn the vortex dynamical system, and accurately predict the evolution of complex vortex structures, such as Taylor and Leapfrog vortices. NSSNN, based on the first principle of learning complex systems, has potential applications in fields of physics, astronomy, and weather forecast, etc. We will further explore the possibilities of neural networks with inherent structure-preserving ability in fields like 3D vortex dynamics and quantum turbulence. In addition, we will also work on general applications of NSSNN with datasets based on images or other real scenes through automatically identifying coordinate variables of Hamiltonian systems based on neural networks.

ACKNOWLEDGMENTS

This project is supported in part by Neukom Institute CompX Faculty Grant, Burke Research Initiation Award, and ByteDance Gift Donation. Yunjin Tong is supported by the Dartmouth Women in Science Project (WISP), Undergraduate Advising and Research Program (UGAR), and Neukom Scholars Program.

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

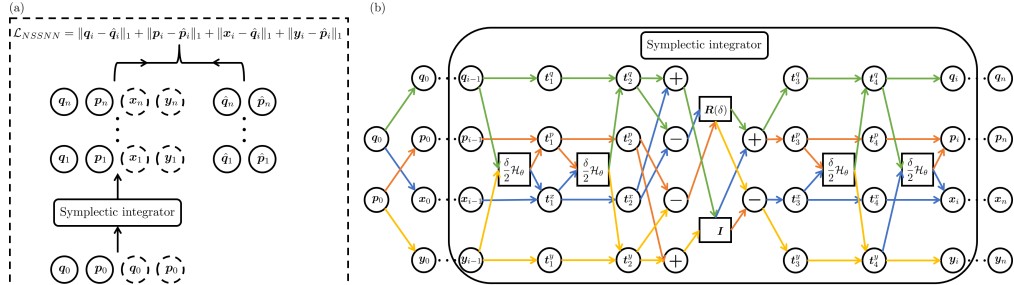

Figure 8: (a) The forward pass of an NSSNN is composed of a forward pass through a differentiable symplectic integrator as well as a backpropagation step through the model. (b) The schematic diagram of NSSNN.

B. Zhu, R. Zhang, Y. Tang, X. Tu, and Y. Zhao. Splitting k-symplectic methods for non-canonical separable hamiltonian problems. *J. Comput. Phys.*, 322:387–399, 2016.

E. E. Zotos. Classifying orbits in the classical hénon–heiles hamiltonian system. *Nonlinear Dynam.*, 79:1665–1677, 2015.

## A   NETWORK ARCHITECTURE

Figure 8(a) shows the forward pass of NSSNN is composed of a forward pass through a differentiable symplectic integrator as well as a backpropagation step through the model. Figure 8(b) plots the schematic diagram of NSSNN. For the constructed network $\mathcal{H}_\theta(\boldsymbol{q}, \boldsymbol{p})$, we integrate (4) by using the second-order symplectic integrator (Tao, 2016). Specifically, The input layer of the integrator is $(\boldsymbol{q}, \boldsymbol{p}, \boldsymbol{x}, \boldsymbol{y}) = (\boldsymbol{q}_0, \boldsymbol{p}_0, \boldsymbol{q}_0, \boldsymbol{p}_0)$ at $t = t_0$ and the output layer is $(\boldsymbol{q}, \boldsymbol{p}, \boldsymbol{x}, \boldsymbol{y}) = (\boldsymbol{q}_n, \boldsymbol{p}_n, \boldsymbol{x}_n, \boldsymbol{y}_n)$ at $t = t_0 + n\mathrm{d}t$. The recursive relations of $(\boldsymbol{q}_i, \boldsymbol{p}_i, \boldsymbol{x}_i, \boldsymbol{y}_i), i = 1, 2, \cdots, n$, are expressed by the algorithm 1.

## B   SYMPLECTOMORPHISMS

One of the most important features of the time evolution of Hamilton's equations is that it is a symplectomorphism, representing a transformation of phase space that is volume-preserving. In the setting of canonical coordinates, symplectomorphism means the transformation of the phase flow of a Hamiltonian system conserves the symplectic two-form

$$\mathrm{d}\boldsymbol{q} \wedge \mathrm{d}\boldsymbol{p} \equiv \sum_{j=1}^{N} (\mathrm{d}q_j \wedge \mathrm{d}p_j), \tag{11}$$

where $\wedge$ denotes the wedge product of two differential forms. The rules of wedge products can be found in Lee (2010). In the two-dimensional case, (11) can be understood as the area element of the surface. In this case, the symplectomorphism can be interpreted as the area element of the surface is constant. As proved below, our constructed network structure intrinsically preserves Hamiltonian structure.

**Theorem B.1.** *For a given $\delta$, the mapping $\phi_1^\delta$, $\phi_2^\delta$, and $\phi_3^\delta$ in (5) are symplectomorphisms.*

*Proof.* Let

$$(\boldsymbol{t}_j^q, \boldsymbol{t}_j^p, \boldsymbol{t}_j^x, \boldsymbol{t}_j^y) = \phi_j^\delta(\boldsymbol{q}, \boldsymbol{p}, \boldsymbol{x}, \boldsymbol{y}), \;\; j = 1, 2, 3. \tag{12}$$

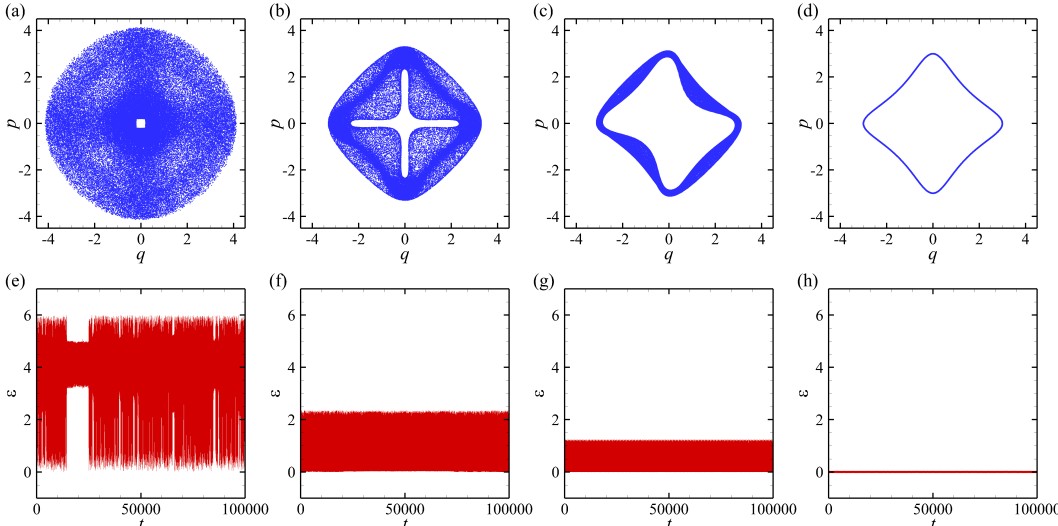

Figure 9: Comparison of different $\omega$ (from left to right columns) with (a) and (e) $\omega = 0$, (b) and (f) $\omega = 0.8$, (c) and (g) $\omega = 0.9$, (d) and (h) $\omega = 10$ in the symplectic integrator for nonseparable Hamiltonian $H = (q^2 + 1)(p^2 + 1)/2$. The upper row: projection of a trace $[q(t), p(t), x(t), y(t)]$ with $[q(0), p(0), x(0), y(0)] = (-3, 0, -3, 0)$ onto the $q - p$ plane. The bottom row: deviation $\epsilon = \|(q, p) - (x, y)\|_2$ between $(q, p)$ and $(x, y)$.

.

From the first equation of (5), we have

$$
\begin{aligned}
&\mathrm{d}\boldsymbol{t}_1^q \wedge \mathrm{d}\boldsymbol{t}_1^p + \mathrm{d}\boldsymbol{t}_1^x \wedge \mathrm{d}\boldsymbol{t}_1^y \\
=&\mathrm{d}\boldsymbol{q} \wedge \mathrm{d}\left[\boldsymbol{p} - \delta \frac{\partial \mathcal{H}_\theta(\boldsymbol{q}, \boldsymbol{y})}{\partial \boldsymbol{q}}\right] + \mathrm{d}\left[\boldsymbol{x} + \delta \frac{\partial \mathcal{H}_\theta(\boldsymbol{q}, \boldsymbol{y})}{\partial \boldsymbol{p}}\right] \wedge \mathrm{d}\boldsymbol{y} \\
=&\mathrm{d}\boldsymbol{q} \wedge \mathrm{d}\boldsymbol{p} + \mathrm{d}\boldsymbol{x} \wedge \mathrm{d}\boldsymbol{y} + \delta \left[\frac{\partial \mathcal{H}_\theta(\boldsymbol{q}, \boldsymbol{y})}{\partial \boldsymbol{q} \partial \boldsymbol{y}} - \frac{\partial \mathcal{H}_\theta(\boldsymbol{q}, \boldsymbol{y})}{\partial \boldsymbol{y} \partial \boldsymbol{q}}\right] \mathrm{d}\boldsymbol{q} \wedge \mathrm{d}\boldsymbol{y} \\
=&\mathrm{d}\boldsymbol{q} \wedge \mathrm{d}\boldsymbol{p} + \mathrm{d}\boldsymbol{x} \wedge \mathrm{d}\boldsymbol{y}.
\end{aligned}
\tag{13}
$$

Similarly, we can prove that $\mathrm{d}\boldsymbol{t}_2^q \wedge \mathrm{d}\boldsymbol{t}_2^p + \mathrm{d}\boldsymbol{t}_2^x \wedge \mathrm{d}\boldsymbol{t}_2^y = \mathrm{d}\boldsymbol{q} \wedge \mathrm{d}\boldsymbol{p} + \mathrm{d}\boldsymbol{x} \wedge \mathrm{d}\boldsymbol{y}$. In addition, from the third equation of (5), we can directly deduce that $\mathrm{d}\boldsymbol{t}_3^q \wedge \mathrm{d}\boldsymbol{t}_3^p + \mathrm{d}\boldsymbol{t}_3^x \wedge \mathrm{d}\boldsymbol{t}_3^y = \mathrm{d}\boldsymbol{q} \wedge \mathrm{d}\boldsymbol{p} + \mathrm{d}\boldsymbol{x} \wedge \mathrm{d}\boldsymbol{y}$. $\qquad \square$

Suppose that $\Phi_1$ and $\Phi_2$ are two symplectomorphisms. Then, it is easy to show that their composite map $\Phi_2 \circ \Phi_1$ is also symplectomorphism due to the chain rule. Thus, the symplectomorphism of algorithm 1 can be guaranteed by the theorem B.1.

## C    DETERMINING COEFFICIENT $\omega$

To further elucidation, the Hamiltonian $\mathcal{H}_A + \mathcal{H}_B$ without the binding, i.e., $\overline{\mathcal{H}}$ with $\omega = 0$, in extended phase space $(\boldsymbol{q}, \boldsymbol{p}, \boldsymbol{x}, \boldsymbol{y})$ may not be integrable, even if $\mathcal{H}(\boldsymbol{q}, \boldsymbol{p})$ is integrable in the original phase space $(\boldsymbol{q}, \boldsymbol{p})$. However, $\mathcal{H}_C$ is integrable. Thus, as $\omega$ increases, a larger proportion in the phase space for $\overline{\mathcal{H}}$ corresponds to regular behaviors (Kolmogorov, 1954). For $H(q, p) = (q^2 + 1)(p^2 + 1)/2$, shown in Fig. 9, we compare the trajectories starting from $[q(0), p(0), x(0), y(0)] = (-3, 0, -3, 0)$ calculated by the symplectic integrator (Tao, 2016) with different $\omega$, where the calculation accuracy is second order accuracy and the time interval is 0.001. As Figs. 9(a), (b), (c), and (d) shown, the chaotic region in phase space is significantly decreasing until forming a stable limit cycle. We define $\epsilon = \|(q, p) - (x, y)\|_2$ as the calculation error of this system, shown in Figs. (e), (f), (g), and (h) that the error is decreasing with $\omega$ increasing, which fits the quantitative results of phase trajectory well.

Table 2: Comparison of Prediciton error and Hamiltonian deviation between NeuralODE, HNN and NSSNN

| | Prediciton error | | | Hamiltonian deviation | | |
|---|---|---|---|---|---|---|
| Problems | NeuralODE | HNN | NSSNN | NeuralODE | HNN | NSSNN |
| Pendulum | $3.4 \times 10^{-2}$ | $3.1 \times 10^{-2}$ | $\mathbf{2.6 \times 10^{-2}}$ | $1.5 \times 10^{-2}$ | $1.3 \times 10^{-2}$ | $\mathbf{7.4 \times 10^{-3}}$ |
| Lotka-Volterra | $\mathbf{2.2 \times 10^{-2}}$ | $3.9 \times 10^{-2}$ | $2.7 \times 10^{-2}$ | $\mathbf{7.4 \times 10^{-3}}$ | $8.8 \times 10^{-3}$ | $7.5 \times 10^{-3}$ |
| Spring | $2.1 \times 10^{-2}$ | $2.1 \times 10^{-2}$ | $\mathbf{1.6 \times 10^{-2}}$ | $9.3 \times 10^{-3}$ | $\mathbf{6.7 \times 10^{-3}}$ | $\mathbf{6.7 \times 10^{-3}}$ |
| Hénon–Heiles | $1.0 \times 10^{-1}$ | $9.4 \times 10^{-2}$ | $\mathbf{8.4 \times 10^{-2}}$ | $3.7 \times 10^{-2}$ | $4.0 \times 10^{-2}$ | $\mathbf{3.5 \times 10^{-2}}$ |
| Tao's example | $3.7 \times 10^{-2}$ | $2.6 \times 10^{-2}$ | $\mathbf{2.2 \times 10^{-2}}$ | $1.4 \times 10^{-2}$ | $1.1 \times 10^{-2}$ | $\mathbf{8.2 \times 10^{-3}}$ |
| Schrödinger | $8.7 \times 10^{-2}$ | $5.9 \times 10^{-2}$ | $\mathbf{5.7 \times 10^{-2}}$ | $3.8 \times 10^{-2}$ | $2.3 \times 10^{-2}$ | $\mathbf{2.0 \times 10^{-2}}$ |
| Vortex (2 particles) | $2.1 \times 10^{-2}$ | $7.7 \times 10^{-3}$ | $\mathbf{3.4 \times 10^{-3}}$ | $1.5 \times 10^{-2}$ | $2.8 \times 10^{-3}$ | $\mathbf{2.1 \times 10^{-3}}$ |
| Vortex (4 particles) | $3.4 \times 10^{-2}$ | $9.4 \times 10^{-3}$ | $\mathbf{6.9 \times 10^{-3}}$ | $8.2 \times 10^{-2}$ | $1.4 \times 10^{-2}$ | $\mathbf{3.4 \times 10^{-3}}$ |

## D   OTHER EXPERIMENTS

We consider the pendulum, the Lotka–Volterra, the Spring, the Hénon–Heiles, the Tao's example (Tao, 2016), the Fourier form of nonlinear Schrödinger and the vortex particle systems in our implementation. The Hamiltonian energies of these systems (except vortex particle system) are summarized as follows:

Pendulum system: $\mathcal{H}(q, p) = 3(1 - \cos(q)) + p^2$. Lotka–Volterra system: $\mathcal{H}(q, p) = p - e^p + 2q - e^q$. Spring system: $\mathcal{H}(q, p) = q^2 + p^2$. Hénon–Heiles system: $\mathcal{H}(q_1, q_2, p_1, p_2) = (p_1^2 + p_2^2)/2 + (q_1^2 + q_2^2) + (q_1^2 q_2 - q_2^3/3)/2$. Tao's example (Tao, 2016): $\mathcal{H}(q, p) = (q^2 + 1)(p^2 + 1)/2$. Fourier form of nonlinear Schrödinger equation: $\mathcal{H}(q_1, q_2, p_1, p_2) = \left[ (q_1^2 + p_1^2)^2 + (q_2^2 + p_2^2)^2 \right]/4 - (q_1^2 q_2^2 + p_1^2 p_2^2 - q_1^2 p_2^2 - p_1^2 q_2^2 + 4 q_1 q_2 p_1 p_2)$.

The network is trained by the dataset with noise $\sim 0.1 U(-1, 1)$. The training time span, integral time step, and validation time span are $0.01$, $0.01$, and $0.1$, respectively. Table 2 compares the Hamiltonian deviation $\epsilon_H = \|\mathcal{H}(\boldsymbol{q}_{\text{truth}}, \boldsymbol{p}_{\text{truth}}) - \mathcal{H}(\boldsymbol{q}_{\text{predict}}, \boldsymbol{p}_{\text{predict}})\|_2 / \|\mathcal{H}(\boldsymbol{q}_{\text{truth}}, \boldsymbol{p}_{\text{truth}})\|_2$ and the prediction error $\epsilon_p = \|\boldsymbol{q}_{\text{truth}} - \boldsymbol{q}_{\text{predict}}\|_1 + \|\boldsymbol{p}_{\text{truth}} - \boldsymbol{p}_{\text{predict}}\|_1$. It is clearly from the Table 2 that NSSNN either outperforms or has similar performances as NeuralODE and HNN do.

