# OpenReview forum: "Nonseparable Symplectic Neural Networks"
_ICLR.cc/2021/Conference — ICLR 2021 Poster_

### Official Review · AnonReviewer1 · 2020-10-26
**Necessary missing component in the Hamiltonian Neural Networks ecosystem.**

**Rating:** 6
**Confidence:** 4

**Review:**

The authors propose a variation of Hamiltonian Neural Networks (HNNs) with a built-in symplectic integrator. While built-in symplectic integrators have already been used in HNNs, the most popular symplectic integrators only work for separable Hamiltonians. The authors use Tao’s integrator (2016) at the core of their model (HSSNN) to achieve better conservation of energy in systems with non-separable Hamiltonians. Many works have experimented with training with different integrators, and in that sense this is a very incremental contribution, however, Tao’s integrator is a state of the art integrator, and I think the community will benefit from reading about it, and seeing it implemented.

The introduction and model description part of the paper is clear in general and reasonably easy to follow, and does a very decent job at introducing Tao’s integrators in a very compact way. Many details for reproducibility are missing from the experimental sections of the paper (specially with respect to the implementation of the baselines), but this should be alleviated by the source code provided by the authors. The final section on the multi-particle vortices is probably the most unclear, and should be improved.

Considering that this is a very empirical paper, with an incremental contributions that only make use of toy environments, it should probably do a bit more of a thorough job at really trying to ablate the model in more sophisticated ways.

I think there are two important baselines missing:
* Performance of their model e.g. using a RK4 integrator, instead of Tao's. This would really tease apart the the effect of the symplecticity. Note this is an important ablation that is between this model and HNN (which is trained by supervising the gradients, rather than through an integrator).
* Training HNN using the gradients, but then integrate the learned Hamiltonian using Tao’s integrator at test time (or even better: a time adaptive version of Tao’s integrator). I would be surprised if this does not perform as well as HSSNN (Although as the authors say, requiring the gradients to train HNN is a very rigid constraint, so this would not take away from the importance of their work).

Also in the multi-particle system, I am surprised the authors did not choose a Hamiltonian Graph Network (https://arxiv.org/pdf/1909.12790.pdf). I do not think at this stage I would need to see this included, but probably a mention to Hamiltonian Graph Networks in the context of multi-particle systems would be appropriate.

Also, Figure 3 b) and c), seem to have weird patterns that would be interesting to try to gain more insights into because I cannot tell if, for example, the variance in Fig 3c is just noise, or something more intrinsic to the experiment. So I really don’t know what to make of plots Figure 3 b) and c). Maybe plotting them for more environments or adding some form of errors bars would help understand them better.

Similarly, in Table 3, I would have expected larger differences between HNN, and HSSNN for the Hamiltonian Deviation column. Maybe it would be possible to use longer trajectories and amplify the differences more? In fact the differences in terms of conservation of energy for Tao’s environment seem to be much smaller quantitatively in Table 3 than qualitatively in Figure 4, was the example from Figure 4 cherrypicked?

I think some additional comparison ablations, like some of those provided in https://arxiv.org/pdf/1909.12790.pdf, would really help gain more insight on the model:
* Try different integrators at train and test time
* Train on a range of time steps during training, instead of a single values.

I think the paper is a bit borderline, and in its current form a lean on the side of "Weak Reject", but would be happy to raise the score, if the authors can make improvements in the axis mentioned above.

Some questions I am curious about (have not affected my decision):
* Any particular reason to use an l1 loss, instead of l2? I wonder if this can also cause differences with the baselines which use l2. The prediction error seems to be defined also using l1. Is it not unfair to compare l1 to the baselines, when baselines are trained with l2 loss (or at least they were in their original papers)?
* Table 1 says that HNN requires the gradients, but if I remember correctly the HNN paper model does mention the possibility of estimating the derivatives with finite differences, which is equivalent to training through an Euler integrator.

Some minor comments (have not affected my decision):

The text in the experimental and results sections feels a bit rough at times, would recommend rewriting most of it, making sure that the message of each sentence is clear and unambiguous.

> these nonseparable systems exhibit plateau of degrees of freedom, demonstrating complexities that are orders-of-magnitude higher than separable systems (whose degree of Freedoms are typically below 10).

Are not many n-body systems highly separable, yet they have much much higher numbers of degrees of freedom?

Figure 1 could be clearer, the intended message of the crossed-out notation in red is not obvious.

Figure 2b bit unclear. The plot implies n iteration steps, but this is not really conveyed really well.

Missing “/” in line 169
Misspelled “systmes” (197)

Notation in equation (7):
Vectors (bold) or scalars (italics)
What about the subindex, should I assume you train on one step data, and not on sequences?
If the formula refers to the training loss, the limit of the sum should probably be the batch size, and not the number of training samples?

The term “Strong stability” in Table 1 seems not very scientific. Maybe something like “symplectic stability” or something like that would be more specific.

Table 2: Hamiltonian Deviation (Row “spring”) HNN and NSSNN seem to have the same error, so maybe both should be in bold.

Equation 2016 wrong left side, H(p, q)_predict, should probably be H(p_predict, q_predict) since I assume the analytical Hamiltonian formula is used in all cases, to estimate the energy of the state.

MODELING VORTEX DYNAMICS OF MULTI-PARTICLE SYSTEM section needs more clarity. For example notation in equations 8 and 9 seems inconsistent. Also lines 229 to 232 lack sufficient detail of how the generalization from 2 particles to N particles is achieved. My guess is that the authors just add up all possible pairwise interactions when moving towards systems with higher number of particles.

EDIT: Updated rating after author revisions.

---

> ### Author Response · Authors · 2020-11-24
> **Response to reviewer 1**
>
> Thank you for your valuable comments. Please see our responses as below:
>
> 1. More descriptions on the experimental section and the N-body problem.
>
> See MAIN UPDATES 4 and 5.
>
> 2. Ablation test on the integrator
>
> See MAIN UPDATES 3. The main conclusion we drew from the ablation test is Tao's integrator is critical when we embed it in HNN and NSSNN, since it effectively avoids error accumulation from the numerical calculation.
>
> 3. Hamiltonian Graph Network
>
> In the multi-particle system, we build a graph neural network similar to the Hamiltonian Graph Network. We cited this paper in the related works and Section 6.
>
>
> 4. Figure 3 b) and c)
>
> See MAIN UPDATES 2. We rewrite this part in Section 4.
>
> 5. Table 3 and Figure 4
>
> As for the error comparison in Table 3, it only shows significant difference after a long-term calculation (like 10000 time steps), Table 3 does not effectively show the difference between HNN and HSSNN. Thus, We move Table 3 to Appendix D and add Figure 5 and 6 with the predicting period $T_{predict}$ up to 20,000. We also redesigned the experiments for comparison to ensure fairness (See Section 5.2).
>
> 6. $L1$ loss vs. MSE loss
>
> We did an ablation test on $L1$ loss vs. MSE loss in Section 4. We found the prediction results using $L1$ loss to train the neural networks are better than that using MSE loss.
>
> 7. Degree of freedoms
>
> We delete the ambiguous sentence "whose degree of Freedoms are typically below 10". It is correct that many n-body systems are highly separable, yet they have higher numbers of degrees of freedom.
>
>
> 8. N-body systems
>
> See MAIN UPDATES 5.
>
>
> 9. In the revised edition, we rewrite most parts that are ambiguous and adopt your other valuable suggestions as well.

---

> > ### Comment · AnonReviewer1 · 2020-11-24
> > **Reviewer response**
> >
> > Thank your for the responses and additional ablations, I am reasonably satisfied, and have increased my rating accordingly.
> >
> > Some suggestions/final questions:
> > * Very interesting to see that L1 vs MSE plays such an important role when training with noise.
> > * Interesting how the IHNN fails in Fig 6, with a very sharp orbit, but shifted by an angle. I think this may be because of the estimation of the gradients using the finite difference. I think if you were to train IHNN through Runge Kutta 4 (instead of estimating the gradients with the finite difference, which is equivalent to training through RK1, i.e. Euler), then IHNN would have actually learned to estimate better gradients, and may have been more competitive. (Similarly training IHNN with the true gradients, but that would require access to the gradients which it is ok not to want to assume).
> > * I think some of the training curves would be better visualized as bar plots of the performance after convergence. (Maybe leave the training curves for the supplementary). I would also make the discussion cleaner.
> > * Comma missing in Figure 4 caption (Between 0.01 and 0.1), and in general probably worth doing a general clean up on all of the new text.
> > * Table 1 is nice, but it is more a prior of what the models can do instead of evidence. Since it is in the results section, it may be nice to have some quantitative numbers to support the comparisons in the main text.

---

### Official Review · AnonReviewer4 · 2020-10-28
**Solid work on modeling nonseparable Hamiltonian dynamics, but needs more clarification**

**Rating:** 6
**Confidence:** 3

**Review:**

The paper extends the symplectic family of network architectures towards modeling nonseparable Hamiltonian dynamic systems. More specifically, the paper implements a  symplectic integration schema (from Tao (2016)) for solving arbitrary nonseparable (and separable) Hamiltonian systems within a symplectic neural network architecture. The results from several modeling tasks show that the proposed Nonseparable Symplectic NNs (NSSNNs) are more robust and accurate than vanilla HNNs and NeuralODEs when applied to nonseparable Hamiltonian systems.

Although the idea of modeling nonseparable Hamiltonian systems with Symplectic NNs was already briefly outlined in the SRNN paper (Chen et al 2020), this paper implements it and further analyses various properties of this approach. Overall, the paper is well structured and well written, however there are still some inconsistencies that need to be addressed and clarified.

Namely, the related work discussion is somewhat handled poorly: For instance, the authors state in only one sentence that NSSNNs are closely related to SympNets (Jin et al 2020), without discussing any further details on how are they related and, more importantly, how they differ. Moreover, from that point on, SympNets are never considered (in the experiments) nor mentioned, even though SympNets are indeed able to model nonseparable Hamiltonian systems. In Table 1, that compares the properties of NSSNNs w.r.t some benchmarks, the authors discus "TaylorNet" and "SSINN" - these two are never introduced before. I assume the former refers to Thong et al. 2020, but I have no idea about the latter.

Regarding the choice of \omega, the authors provide some evidence that the choice of \omega plays a role as a regularization, where larger values tend to restrain the system. The analyses given in Appendix B show that with \omega 10 the system already is stable (which also supports the experiments presented in Tao 2016). But then the \omega is set to 2000 in the experiments, which is orders of magnitudes larger than the analyses. How and why was this value chosen?

Lines 206-207 state that from the results in Fig4, it is "clear that" NSSNNs can perform long-term predictions but HNNs and NeuralODEs (in the legend they are listed as ODE-nets, are these the same method?)  fail. It is not clear how was this determined, since the results show that NSSNNs are more robust to noise than the other two, NeuralODEs are still able to perform long-term predictions (in a noiseless setting), and HNNs in a both scenarios w/o noise and w/ moderate amount of noise.

Some typos and minor comments:
	L1: Hamiltonian systems are not a "special" category of physical systems, but is a formalism for modeling certain physical systems (eg. a pendulum, besides within Hamiltonian mechanics, can be modeled within classical (Newtonian) mechanics and Lagrangian mechanics).
	L42: "e.g. see Tong et al. 2020" -> "Tong et al. 2020"
	L56: "degree of Freedoms" -> "degrees of freedom"
	L206: "figure 4" -> "Figure 4"

#Update

I thank the authors for addressing my questions and revising the manuscript, which clarified many of my concerns regarding this work.

---

> ### Author Response · Authors · 2020-11-24
> **Response to reviewer 4**
>
> Thank you for your valuable comments. Please see our responses as below:
>
> 1. Related works
>
> See MAIN UPDATES 1.
>
> 2. Choice of $\omega$
>
> The choice of $\omega$ is discussed in detail in Section 4. In short, the choice of $\omega$ is largely flexible, since NSSNN is not sensitive to the parameter $\omega$ when it is larger than a certain threshold.
>
> 3. Experiments description
>
> See MAIN UPDATES 4.
>
> 4. Some typos and minor comment
>
> We revised all the typos and adopted all the minor comments mentioned.

---

### Official Review · AnonReviewer3 · 2020-10-28
**Reviewer 3**

**Rating:** 6
**Confidence:** 3

**Review:**

The work proposes a novel method for solving non-separable Hamiltonian systems, using Tao's approach in which two copies of the phase space are tied together by an additional Hamiltonian. This appears to be a novel proposal, and certainly of interest.

# Positioning w.r.t. related work.
The proposed method specifically focuses on non-separable systems. The work by Jin et al. (2020) is mentioned in passing, but this work appears to address the problem of non-separability as well. It would thus make sense to dive deeper into how the proposed method compares to Jin's, both theoretically and empirically. Some arguments require citations, such as "variational methods are not well suited for tackling such challenges due to heir inherent weaknesses": which weaknesses?

# Discussion of the method.
Although the method appears to perform favourably in all provided benchmarks, it remains unclear how the method compares on computational and memory complexity. This should be added to the discussion, with a theoretical and/or empirical analysis. It would also be beneficial if areas would be highlighted where the method falls short.

# Experiment section.
Although there is a section called 'Ablation test', an actual empirical ablation study is missing. As the proposed method introduces various moving parts including a specific loss, phase space parametrisation, and specific neural architecture fo H. It is not clear what the relative contribution is of these parts to the improved performance over the baselines. A closer study on this, as well as the influence of the hyperparameters such as t, would shine more light on the characteristics of the proposed method. Moreover, the experimentation section does not convince that the baselines' hyperparameters have been fairly tuned, and if a potential increase of parameters in H might contribute to the stronger performance.

# Clarity of writing.
I found the paper difficult to follow. The meaning of some phrases is unclear, e.g. 'which contributes robust for wider datasets'.

# Conclusion.
The work is interesting, but the analysis of the method is incomplete due to the lack of comparison with a recent competing method, and a missing discussion of the potential tradeoffs involved in choosing this method over others.

Couple nitpicks outside of the review:
- "Nonsep_e_rable" appears a couple of times
- Line 197: 'systmes.

Update: The authors have addressed the most pressing issues with the manuscript. I've increased my score and vote in favour of accept. The section of limitations was difficult to follow and would benefit from a more structured comparison with competing methods.

---

> ### Author Response · Authors · 2020-11-24
> **Response to reviewer 3**
>
> Thank you for your valuable comments. Please see our responses as below:
>
> 1. Compare with Jin et al. (2020)
>
> See MAIN UPDATES 1.
>
> 2. Discussion of the method
>
> See MAIN UPDATES 1 and 6. Section 5.1 discusses the advantages of NSSNN comparing to other methods. Section 7 adds analysis of NSSNN's computational complexity and limitations.
>
> 3. Ablation test
>
> See MAIN UPDATES 2 and 3.
>
> 4. Clarity of writing
>
> In the revised edition, we rewrite most parts that are ambiguous, and adopt your other valuable suggestions as well.
>
> 5. Comparison with a recent competing method
>
> See MAIN UPDATES 1.
>
> 6. Potential tradeoffs
>
> See MAIN UPDATES 6.

---

### Official Review · AnonReviewer2 · 2020-11-02
**recommend accept**

**Rating:** 7
**Confidence:** 3

**Review:**

Summary:
This paper describes a deep learning approach for predicting Hamiltonian systems. The original paper enforces conservation in the loss function. Several of the follow-up papers embed a symplectic integrator instead, but these couldn't handle non-separable systems. This paper can both handle non-separable systems and use a symplectic integrator to enforce conservation. They demonstrate their system on quite a few examples and show lower error (both in prediction and the deviation in the Hamiltonian) than a NeuralODE or the original HNN paper. The final example, in Figure 5, shows a compelling visual improvement.

Strong points:

The Greydanus, et al. NeurIPS 2019 paper on Hamiltonian Neural Networks was very successful and has already inspired many follow-up papers. This paper improves it in several ways (Table 1). Since the ICLR community is similar to the NeurIPS community, I think that this paper would be of interest.

I like that the extension for non-separable systems directly builds off an approach for extending integrators to non-separable systems (Tao, 2016). Further, the Tao integrator is built into the network training. This suggests that it's a robust path to take.

I appreciate that results are reported on quite a few examples to compare NeuralODE, HNN, and the new method (NSSNN).

The results in Figure 5 are quite impressive! I also think it's cool that the training data only needed to be from two particles.

Weak points/clarification questions:

In the examples in this paper, the canonical coordinates need to be known ahead of time. The original HNN paper has an example where the coordinates can be learned from data (Pixel Pendulum), and I believe some of the follow-up papers have covered this case as well. Do you have any thoughts on if your method could learn the coordinates?

The font size in the figures is sometimes hard to read.

In Section 4.2 & Table 2, is there a held-out test set, or are these all training errors? I would like to see both training & test errors to see if there is overfitting.

Minor points:

I found it confusing that Figure 3 & Section 4.1 refer to an "ablation test." It seems like a test to choose a suitable training set.

Disclaimer:

I should mention that I can't vouch for the proof in the appendix.

---

> ### Author Response · Authors · 2020-11-24
> **Response to reviewer 2**
>
> Thank you for your valuable comments. Please see our responses as below:
>
> 1. Learn the coordinates
>
> It is a meaningful project to learn Hamiltonian systems based on images, which mainly relies on an encoded network transforming image data to coordinates data, and then a decoded network transforming coordinates data back to image data. We have covered this part in Section 8 as our future works.
>
> 2. Font size in the figures
>
> We adjust the fonts in pictures to make them more distinguishable.
>
> 3. Overfitting problem
>
> Figure 2 shows the evolution of training loss and validation loss with various conditions, in which it doesn't show obvious overfitting.
>
> 4. Ablation tests
>
> See MAIN UPDATES 2.
>
> 5. Proof in the appendix
>
> We made more explanations on mathematical demonstrations in Appendix B.

---

### Author Response · Authors · 2020-11-24
**MAIN UPDATES**

We thank the reviewer for valuable comments and feedback. We have addressed all the comments in our updated manuscript and marked the revised part in blue color.

MAIN CONTRIBUTIONS:

Compared with other methods, NSSNN makes a methodological improvement in predicting large-scale, multi-particle, and nonseparable Hamiltonian systems without accessing to the time derivatives. As far as we know, none of the existing methods can handle nonseparable Hamiltonian systems on the level of thousands of particles (the SOTA is around 10) and governed by complicated physics (vortical fluid equations).

COMPARISON WITH OTHERS:

SymNets [Jin et al., 2020]: Compared with SymNets, our model incorporates the pairwise interaction prior (with O(1) complexity, the same as the Interaction Network), enabling the process of N-body systems with many DoFs. In contrast, SymNets constructs a series of symplectic mappings composed of matrices with fixed forms in a fully-connected layer (with $O(N^2)$ complexity), which is not scalable for large-scale Hamiltonian systems such as vortical flow (see Section 5.1).

HNNs [Greydanus et al., 2019]: HNN requires the temporal derivative for training, which could be either difficult to obtain or inaccurate due to the large time interval. Our method does not need such derivative information (see the new experiments in Section 5.2).

MAIN UPDATES:

1. Rewrite the related works in Section 2 and the comparisons with other existing methods in Section 5.1, in which we emphasize our methodology improvements.

2. A series of ablation tests are added in Section 4, such as the solver's parameter $\omega$, loss function, integration time steps, and time span of the dataset to evaluate the effect on the network convergence.

3. We also implement ablation tests on different time integrators including replacing Tao's integrator with Runge–Kutta integrator and replacing HNN's integrator with Tao's integrator (see Section 5.2).

4. We rewrite the experimental results in Section 5.2. The parameter settings, the loss function, and the neural network architectures are kept the same during the comparisons between different methods, which ensures the fairness of the comparisons.

5. More detailed descriptions of N-body problem implementations are added in Section 6. We assume the interactive models between particle pairs with unit particle strength are the same, so the corresponding Hamiltonian of N particles can be represented as the sum of the Hamiltonian defined by two-particle pairs.

6. We add Section 7 to describe the limitations. The limitations are mainly the high training cost associated with neural networks with embedded integrators and difficulties in applying NSSNN to real-world systems with large dissipation.

---

### Decision · Program_Chairs · 2021-01-07
**Final Decision**

**Decision:**

Accept (Poster)

**Comment:**

This paper proposes a method for learning non-separable Hamiltonian dynamics by including a state of the art symplectic integrator (Tao, 2016) into the model training pipeline. This is a nice improvement on the past work that primarily addressed separable Hamiltonians. The reviewers agree that the paper is well written and that the empirical evaluation is solid. The paper was further improved during the discussion period by incorporating the reviewers' feedback. For this reason I am happy to recommend this paper for acceptance.